# Effectiveness of a digital quantitative assessment of protein-to-creatinine ratio using computer-vision technologies in proteinuria screening

Jingyi Wu[†]
Advanced Institute of Information
Technology, Peking University
Hangzhou 311215, China
joywu@pku.edu.cn

Yuewen Zheng[†]
Advanced Institute of Information
Technology, Peking University
Hangzhou 311215, China
zhengyuewen27@dingtalk.com

Chunmiao Zhou
Advanced Institute of Information
Technology, Peking University
Hangzhou 311215, China
zhouchunmiao12006@126.com

Fei Wang
Advanced Institute of Information
Technology, Peking University
Hangzhou 311215, China
feiwang1618@dingtalk.com

Qing Li
Advanced Institute of Information
Technology, Peking University
Hangzhou 311215, China
blacknepia@dingtalk.com

Pengfei Li*
Advanced Institute of Information
Technology, Peking University
Hangzhou 311215, China
pfli@bjmu.edu.cn

Luxia Zhang
National Institute of Health Data
Science, Peking University
Beijing 100191, China
zhanglx@bjmu.edu.cn

## ABSTRACT

**Objective**: Proteinuria is an effective indicator for early kidney damage, and is recommended to be tested routinely in population at high risk of chronic kidney disease (CKD). This study proposed a novel digital approach, the quantitative urine protein-to-creatinine ratio (UPCR) home-testing kit (UTK), for proteinuria screening based on advanced computer-vision algorithms, and the screening effectiveness was evaluated.

**Method**: A total of 199 participants who have visited the Peking University First Hospital in October 2023, were included in our study. Randomly selected spot urine samples were collected and measured using laboratory and UTK methods respectively. The UTK method utilized the contour detection and image segmentation algorithms to extract the topological information of the urinalysis strip for color calibration, and employed the three-dimensional color space interpolation algorithm for quantitative readings of UPCR. With the laboratory results as golden criteria, the diagnostic performance of UTK in proteinuria screening was evaluated in terms of validity, reliability, predictive values, and area under the receiver operating characteristic curve (AUC). We conducted the Bland-Altman analysis to assess the agreement of the quantitative UPCR results between UTK and laboratory methods across different levels of UPCR.

**Result**: The mean age of the 199 participants was 51.6 ± 16.2 years and 88 (44.2%) of them were male. The median UPCR was 222.5 mg/g (interquartile range: 106.5-844.6), and 20 (10.1%) participants were identified as proteinuria. The UTK method performed well in proteinuria screening, with a high accuracy of 91.0%, an 85.0% sensitivity, a 91.6% specificity, and a 0.996 AUC. The Bland-Altman plot showed high agreement of the quantitative UPCR results between the UTK and laboratory methods, especially for participants with a relatively low level of UPCR.

**Conclusion**: The proposed digital solution for quantitative UPCR analysis based on advanced computer-vision technologies showed good performance in proteinuria screening. This user-friendly and cost-effective UPCR monitoring method can be a promising new strategy to enhance the efficiency of primary prevention and management of CKD.

## CCS CONCEPTS

• **Applied computing** → **Life and medical sciences** → **Health informatics**

• **Track 2: People-Centric AI Technologies**

## KEYWORDS

Digital health intervention, computer vision, proteinuria, screening, chronic kidney disease

**ACM Reference format:**

Jingyi Wu, Yuewen Zheng, Chunmiao Zhou, Fei Wang, Qing Li, Pengfei Li and Luxia Zhang. 2024. Effectiveness of a digital quantitative assessment of protein-to-creatinine ratio using computer-vision technologies in proteinuria screening. In *KDD 2024 Workshop Artificial Intelligence and Data Science for Healthcare.*

[†]These authors contributed equally to this study.
*Corresponding to Pengfei Li, pfli@bjmu.edu.cn

# 1 INTRODUCTION

As a major public health problem around the world, chronic kidney disease (CKD) has been associated with a range of adverse outcomes, such as end-stage renal disease, cardiovascular disease, and premature death [1]. A national survey has reported that the prevalence of CKD in China is as high as 10.8%, while the awareness rate of CKD is only 10.0% [2]. The rapidly population aging and associated surging prevalence of hypertension and diabetic mellitus are projected to further increase the CKD burden in China. About 73.3% of CKD cases in China are in the early stages, but the lack of apparent symptoms in the early stage of CKD often results in delayed diagnosis and treatment [3, 4]. Previous epidemiological studies have indicated that early diagnosis and timely intervention in the community and primary care settings for CKD are crucial for slowing the disease progression and improving the clinical outcomes for patients [5-7].

Despite the increasing recognition of the heavy burden of CKD, there remains controversy and lack of consensus in terms of the utility of population screening for CKD or targeted screening programs, particularly in light of the associated inconvenience and high costs. Albuminuria or proteinuria are sensitive indicators for early kidney damage and are commonly used in CKD screening. The gold standard for measuring proteinuria has been 24-h urine protein excretion or spot urine protein-to-creatinine (UPCR) using chemistry analyzer in laboratory. For population at high risks of CKD, the Kidney Disease: Improving Global Outcomes (KDIGO) guideline recommends a routine monitoring of indicators of kidney health like proteinuria [8]. However, the international survey addressing global kidney healthcare resources reported that fewer than 1 in 4 surveyed countries had facilities available for routine measurements of proteinuria [9]. To improve cost-efficacy of proteinuria screening, a semiquantitative method based on urine dipstick reagent strips has been proposed in community screening [10]. However, the existing reagent strip devices measuring proteinuria have limited applications due to the insufficient sensitivity and specificity, not adjusting for urinary concentration, and not providing precise measurement values [8]. Given the large population of CKD and the limitations in medical resources, there is a pressing need for a more cost-effective and accurate method for the quantitative measurement of proteinuria.

In this study, based on the advanced computer vision technologies, we have developed a digital quantitative method, the UPCR home-testing kit (UTK), for proteinuria screening. The effectiveness of UTK in proteinuria screening will be evaluated from multiple perspectives, and the promising applications of the digital method for CKD management will be exhibited. This user-friendly and cost-effective method for proteinuria screening can be a novel digital health intervention (DHI) to broaden access to healthcare services among the large population of CKD, especially in under-resourced regions, and to improve the adherence and efficiency of CKD management for a better health outcome.

# 2 METHODS

## 2.1 Participants

A total of 199 participants with various levels of UPCR (ranging from 7.6 to 11449.0 mg/g) were enrolled from the adult patients who visited Peking University First Hospital in October 2023. The sample size was calculated based on the assumed sensitivity of 0.85 and a specificity of 0.85, at a prevalence of proteinuria of 10% and a minimum two-sided 95% confidence interval and 90% power of the study. This research was approved by the Ethics Committee of Peking University First Hospital (No. 2019 [146]), and written informed consent was obtained from all participants.

## 2.2 Identification of proteinuria

For each participant, the random, midstream spot urine specimen was collected, and detection of UPCR were performed using both the automated urine chemistry analyzer in the hospital laboratory and the computer-vision based UTK method. Taking the laboratory results as gold standard, a patient was identified as having proteinuria if his/her UPCR value exceeding 150 mg protein/g creatinine, according to the KDIGO guideline [11].

## 2.3 Computer-vision based UTK

The computer-vision based UTK consists of two parts: a urinalysis reagent strip produced by ACON Biotech (Hangzhou) Co., Ltd. and a standard colorimetric plate for color calibration (**Figure 1**). Based on computer vision technologies, we achieved quantitative reading of urinary test strips and provided a precise UPCR value quickly for each sample. The computer vision algorithms can be briefed as follows. First, a photograph of the strip after urinalysis reaction and the standard colorimetric plate was taken, and then was converted to grayscale, followed by applying median filtering with a kernel size of 7x7 for denoising. After denoising, the Sauvola adaptive thresholding algorithm was used to binarize the image, resulting in a binary image. Second, the contour detection and image segmentation algorithms were utilized to extract the topological information from the binary image for perspective transformation. Third, color calibration of the strip is based on a standard 24-color ColorChecker. Using the least squares method, the color correction matrix (CCM) is iteratively derived, reflecting the mapping between standard and captured 24-color values. The CCM is then applied to calibrate the test strip and standard color patches, obtaining calibrated color values. Based on a three-dimensional color space interpolation algorithm, the quantitative readings of the values of proteinuria and creatinine were obtained from the strip after color calibration, and a precise UPCR was calculated based on the quantitative proteinuria and creatinine measurements.

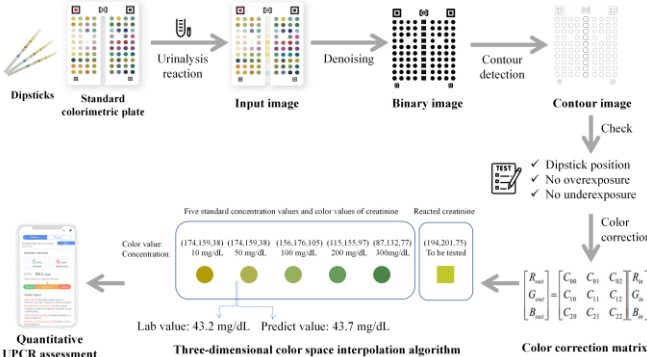

**Figure 1. The UPCR home-testing kit (UTK) based on computer-vision technologies.**

## 2.4 Statistical analysis

For the descriptive analysis, the mean value $\pm$ standard deviation was presented for normally distributed continuous variables and the median value with interquartile range (IQR) for non-normally distributed continuous variables. Count data was demonstrated as number (proportion).

Using the laboratory result as the gold standard, the effectiveness for proteinuria screening of the computer vision-based UTK method was evaluated from four dimensions: the overall performance based on the receiver operating characteristic (ROC) curve, validity, reliability, and predictive values. The validity was measured using sensitivity, specificity, and the Youden Index; the reliability was assessed using the Kappa value and accuracy; and the positive and negative predictive values (PPV and NPV) were calculated as well. Additionally, we evaluated the agreement of UPCR values between the UTK and laboratory methods using the linear regression and the modified Bland-Altman analysis. Subgroup analyses were performed to test whether the screening performance of UTK is robust across different age and sex groups. All statistical analyses in this study were conducted using Python (version 3.8.5, Python Software Foundation, Beaverton, Oregon, USA).

## 3 RESULTS

### 3.1 Population characteristics

In this study, a total of 199 participants were enrolled, with an average age of 51.6 $\pm$ 16.2 years, and 88 (44.2%) of them were male. The participants have a median UPCR of 222.5 mg/g (IQR: 106.5-844.6 mg/g), and 20 (10.1%) of them were identified as proteinuria. The median protein of the studied population was 17.4 mg/dL (IQR: 9.2-48.4 mg/dL), and the median creatinine was 67.8 mg/dL (IQR: 49.9-103.9 mg/dL). The basic characteristics of the study population are detailed in **Table 1**.

**Table 1. Basic characteristics of study population.**

|  | Laboratory | UTK |
|---|---|---|
| Age, years | 51.6±16.2 | |
| Male | 88 (44.2%) | |
| Total protein (mg/dL) | 17.4 (9.2-48.4) | 16.0 (10.3-74.7) |
| Creatinine (mg/dL) | 67.8 (49.9-103.9) | 62.5 (45.0-84.3) |
| UPCR (mg/g) | 222.5 (106.5-844.6) | 321.1 (175.4-1327.2) |
| Proteinuria | | |
| Positive | 20 (10.1%) | 32 (16.1%) |
| Negative | 179 (89.9%) | 167(83.9%) |

### 3.2 Screening performance for proteinuria

As shown in **Table 2** and **Figure 2**, the UTK method performed well in proteinuria screening in terms of overall performance, validity, reliability, and predictive values. Specifically, the UTK method achieved a remarkable overall performance with a high area under the ROC curve (AUC) of 0.996. In terms of validity, this method also yielded a high sensitivity of 85.0, a high specificity of 91.6, and a 76.6% Youden index. In terms of reliability, UTK achieved a considerable accuracy of 91.0%, and the Kappa value was 0.60. As for the predictive values, we observed a high NPV of 98.2% and a relatively limited PPV of 53.1%.

The linear regression analysis and the Bland-Altman plot were utilized to evaluate the agreement between UPCRs measured by UTK and laboratory methods. The scatter plot between UPCRs measured by UTK and laboratory methods was shown in **Figure 3**, and the $R^2$ from the linear regression analysis was 0.558 for the UTK method, suggesting that the UPCRs measured using UTK were generally comparable to laboratory values. Compared with the laboratory values, the UPCR measured by UTK was slightly lower among the population with a relatively high UPCR.

With most of the points falling within the 95% confidence interval of bias, the Bland-Altman plot (**Figure 4**) suggested a strong consistency between UPCR measured by UTK and laboratory methods. For participants with a lower level of UPCR, the UTK method performed better with less bias, namely, the mean difference between UPCR measured by the UTK and laboratory methods was smaller.

**Table 2. Screening performance for proteinuria.**

| Dimension | Indicator | Value |
|---|---|---|
| **Validity** | Sensitivity | 85.0% |
| | Specificity | 91.6% |
| | Youden index | 76.6% |
| **Reliability** | Accuracy | 91.0% |
| | Kappa value | 0.60 |
| **Predictive values** | Positive predictive value | 53.1% |
| | Negative predictive value | 98.2% |

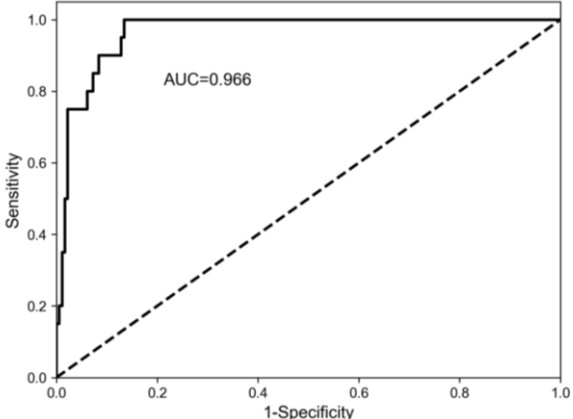

**Figure 2. The receiver operating characteristic curve for proteinuria screening using UTK.**

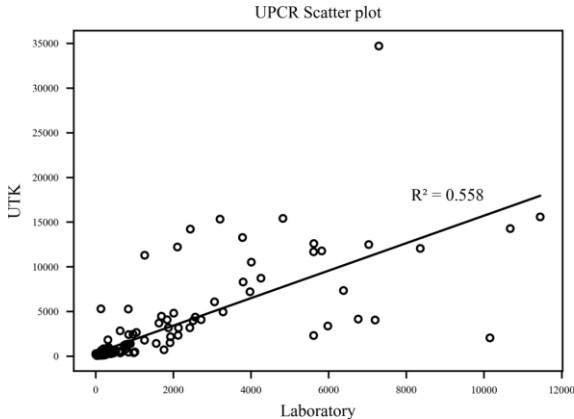

**Figure 3. Linear correlation of UPCR between UTK and laboratory methods.**

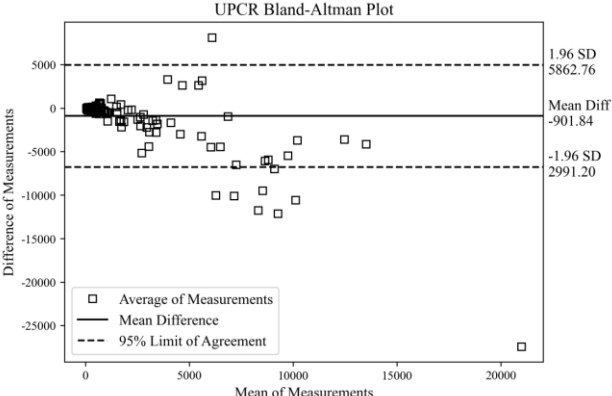

**Figure 4. Bland-Altman plot of the agreement of UPCR between UTK and laboratory methods.**

## 3.3 Subpopulation analysis

Generally, the effectiveness of UTK for proteinuria screening was robust across different subpopulation stratified by age and sex (**Table 3**), with a high accuracy ranging from 88.2% to 94.4%, a sensitivity ranging from 77.8% to 90.9%, and a specificity ranging from 88.3% to 94.9%. Specifically, among the participants aged <50 years old, the UTK method yielded a slightly better performance in screening, with an accuracy of 94.4%, a sensitivity of 90.0%, and a specificity of 94.9%. As for subpopulation stratified by sex, a slightly higher accuracy (92.8%) and specificity (94.1%), but a slightly lower specificity (77.8%) of UTK was observed for the female population.

**Table 3. Screening performance for proteinuria by subpopulation.**

|  | N | UPCR (mg/g) | Accuracy | Sensitivity | Specificity |
|---|---|---|---|---|---|
| **Age** |  |  |  |  |  |
| < 50 | 110 | 150.4 (81.4-349.5) | 94.4% | 90.0% | 94.9% |
| ≥ 50 | 89 | 305.6 (147.7-1007.8) | 88.2% | 80.0% | 89.0% |
| **Sex** |  |  |  |  |  |
| Male | 88 | 205.6 (99.8-1105.5) | 88.6% | 90.9% | 88.3% |
| Female | 111 | 242.2 (110.1-741.8) | 92.8% | 77.8% | 94.1% |

## 4 DISCUSSION

This study has proposed a novel digital approach for cost-effective proteinuria screening and kidney health self-monitoring for population at high risk of CKD progression. Based on data of 199 participants with the range of UPCR from 7.6 to 11449.0 mg/g, we found that the computer vision-based UPCR home-testing method performed well in proteinuria screening with a high accuracy, sensitivity, and specificity. This user-friendly digital method for kidney health monitoring can help improve the awareness and self-management of kidney diseases among the large-scale Chinese population, and can be a promising strategy for primary prevention of CKD, potentially alleviating the heavy burden on healthcare systems of kidney disease in China.

The aging population has led to a significant increase in prevalence of metabolic diseases such as obesity, hypertension, and diabetic mellitus, which are major risk factors for kidney disease and can contribute to a substantially growing burden of CKD in China [12-15]. Kidney disease often goes undetected due to a lack of early symptoms and low awareness; however, many epidemiological evidences have suggested that early diagnosis and intervention of CKD are associated with declined disease progression and better health outcomes [5-7]. The absence of symptoms of CKD in early stages makes laboratory testing

imperative for at-risk patients, and the disease progression can be in-time influenced by public health measures, lifestyle interventions, and inexpensive drug treatments [16]. The KDIGO guideline recommends a routine monitoring of biomarkers indicative of kidney damage in high-risk population of CKD [8], and in clinical practice, proteinuria or albuminuria quantified in random single-voided urine specimens are commonly used to monitor kidney injuries. The gold standard for proteinuria screening is the quantitative method utilized in the clinical laboratories, which faces challenges related to patient adherence due to its relatively high economic cost and time requirements. Alternatively, a semi-quantitative method based on urine dipstick reagent strips is proposed for large-scale screening of proteinuria [17, 18]. While this approach is relatively cost-effective, the clinical utility of these dipstick measurements is limited due to a low accuracy [18]. For instance, a commercially available dipstick test of Siemens, Indiana, USA, the Clinitek Atlas®PRO™12 Reagent Pak, was reported with a sensitivity of 70.0 % and a specificity of 95.9 % in proteinuria screening among 2,932 samples in Taiwan, China [17]. Another UPCR urinalysis dipstick test in Ghana, the Life Assay Diagnostics (LAD) Test-it™, showed a sensitivity of 50.7% and a specificity of 69.2% in detection of proteinuria in a representative antenatal care setting [18]. Most of existing semi-quantitative methods for proteinuria based on urine dipsticks rely on inaccurate manual readings or complex reading devices like reflectance meters, and thus are not user-friendly for home testing and self-monitoring of kidney health. Leveraging on cutting-edge computer vision technologies, UTK automated the interpretation of dipstick-based urinalysis by mobile applications, and could provide quantitative measurements for 12 indicators of kidney health, including urine albumin-to-creatinine ratio (UACR) and UPCR. Our previous study demonstrated that UTK exhibited superior performance in albuminuria screening from multiple perspectives based on quantitative UACR measurements [19]. This study further contributes to the evidence that UTK achieves high consistency and robust performance in quantitative UPCR measurement and proteinuria screening. The critical role of proteinuria and albuminuria in CKD detection and management is well established, and this digital urinalysis method can be a promising new strategy for CKD screening and self-management in China, particularly in light of the increasing burden of CKD on healthcare systems.

DHIs, which leverage digital tools to enhance healthcare delivery, have shown effectiveness in managing CKD and improving the quality, safety, and efficiency of primary care [20]. World Health Organization has acknowledged that we must learn to harness the power of DHIs and digital technologies more generally to improve healthcare delivery for all [21]. Based on our assessments, UTK might be a user-friendly and cost-effective DHI that can be integrated into existing hierarchical systems of CKD management in China (**Figure 5**), especially for supporting primary care in under-resourced healthcare settings. Specifically, in CKD surveillance, UTK can be utilized in primary CKD screening among population with risk factors (e.g., hypertension and diabetic mellitus), and those testing positive by UTK will be transferred to the primary care units for further diagnostic

evaluation. For management of patients with mild to moderate CKD, UTK can be used for routine monitoring of kidney health, and patients can achieve self-management under the support of primary care units. To ensure the last translational bridge to implementation is crossed, future studies are expected to demonstrate cost-effectiveness, sustainability plans, beneficial effects on quality of life measures and reductions in healthcare utilization costs for UTK applications.

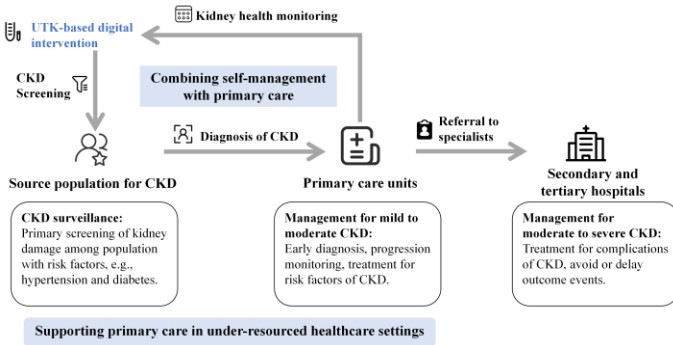

**Figure 5**. **Application of UTK in hierarchical management of CKD**.

Our study has several limitations. First, this is a single-center study based on a relatively small sample, and the generalizability of our results need to be validated in a large-scale multicenter population. Second, due to the limitations of dipstick-based urinalysis, the maximum detectable level of urine protein is 2000 mg/dL for UTK. Third, personal information regarding the baseline health status, medication use, dietary and so on, were not available in this study, which might influence our assessments. Additionally, the performance of UTK can be further enhanced by optimization of the computer-vision algorithms, and a cost-effectiveness analysis based on real-world data should be conducted for this digital solution in enhancing CKD management in clinical practice.

In conclusion, the present study proposed a digital solution of quantitative UPCR analysis based on advanced computer-vision technologies, which has been evaluated with good performance in proteinuria screening. This user-friendly and cost-effective UPCR monitoring method can be a promising new strategy to enhance the efficiency of CKD management and address the increasing needs of the large-scale CKD population in China.

## ACKNOWLEDGMENTS

This work was supported by the Ministry of Science and Technology of the People's Republic of China (2022YFF1203001).

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
