# OpenReview forum: "Effectiveness of a digital quantitative assessment of protein-to-creatinine ratio using computer-vision technologies in proteinuria screening"
_KDD.org/2024/Workshop/AIDSH — KDD-AIDSH 2024 Oral_

### Official Review · Reviewer_J72F · 2024-06-11
**The authors propose a digital solution for quantitative urine protein-to-creatinine ratio analysis based on computer-vision technologies, which shows good performance in proteinuria screening on an in-house cohort consisting of 199 participants. The paper is well written and easy to understood. The used computer-vision technologies are basic, but need further description.**

**Rating:** 7
**Confidence:** 4

**Review:**

The paper has a clinical significant objective and present a digital solution. The paper is well written and easy to understood. The used computer-vision technologies are basic, but need further description.

1.what denosing algorithms are used? How is the color correction matrix generated?

---

### Official Review · Reviewer_okuH · 2024-06-17
**Evaluation of "Effectiveness of a digital quantitative assessment of protein-to-creatinine ratio using computer-vision technologies in proteinuria screening"**

**Rating:** 6
**Confidence:** 4

**Review:**

This paper proposed an innovative digital approach for proteinuria screening using a quantitative urine protein-to-creatinine ratio home-testing kit based on computer-vision technologies. The authors provide a detailed description of study design. The use of advanced computer vision algorithm for quantitative UPCR is a novel. My major comments are:
1.	The sample size is a little small. This may limit the statistical power of the findings. Conducting subgroup analysis may result in a smaller sample size.
2.	Can you consider more health information The lack of health information may influence the results and can be considered in future study.
Overall, the paper provides a promising new strategy for chronic kidney disease.

---

### Decision · Program_Chairs · 2024-06-28

Accept (Oral)